# Polar Motion Ultra-Short-Term Prediction of Least-Squares+Multivariate Autoregressive Hybrid Method by Using the Kalman Filter

**DOI:** 10.3390/s24196260

**Published:** 2024-09-27

**Authors:** Zhirong Tan, Fei Ye, Liangchun Hua

**Affiliations:** 1Bei Dou High-Precision Satellite Navigation and Location Service Hunan Engineering Research Center, Hunan Institue of Geomatics Sciences and Technology, Shaoshanzhong Road No. 693, Changsha 410007, China; 2Yi Yang Satellite Application Technology Center, Yi Yang Natural Resources and Planning Bureau, Longzhou South Road No. 299, Yiyang 413000, China

**Keywords:** LS+MAR, LS+AR, Kalman filter, ultra-short-term prediction, polar motion

## Abstract

The polar motion (PM, including two parameters PMx and PMy) ultra-short-term prediction (1–10 days) is demanded in the real-time navigation of satellites and spacecrafts. Improving the PMx and PMy ultra-short-term predictions accuracies are a key to optimize the performance of these related applications. Currently, the least squares (LS)+autoregressive (AR) hybrid method is regarded as one of the most capable approaches for ultra-short-term predictions of PMx and PMy. The Kalman filter has proven to be effective in improving the ultra-short-term prediction performance of the LS+AR hybrid method, but the PMx and PMy ultra-short-term predictions accuracies are still not able to satisfy some related applications. In order to improve the performance of PM ultra-short-term prediction, it is worth exploring the combinations of existing methods. Throughout the existing predicted methods, the LS+multivariate autoregressive (MAR) hybrid method by using the Kalman filter has the potential to improve the accuracy of PM ultra-short-term prediction. In addition, a PM prediction performance analysis of the LS+MAR hybrid method by using the Kalman filter, namely the LS+MAR+Kalman hybrid method, is still missing. In this contribution, we proposed the LS+MAR+Kalman hybrid method for PM ultra-short-term prediction. The data sets for PM predictions, which range from 1 to 10 days, have been tested based on the International Earth Rotation and Reference Systems Service Earth Orientation Parameter (IERS EOP) 14 C04 series to assess the performance of the LS+MAR+Kalman hybrid model. The experimental results illustrated that the LS+MAR+Kalman hybrid method can effectively execute PMy ultra-short-term predictions. The improvement of PMy prediction accuracy can rise up to 12.69% for 10-day predictions, and the improvement of ultra-short-term predictions is 7.64% on average.

## 1. Introduction

Polar motion (PM, including two parameters PMx and PMy), describes the position variation between the rotational pole and the crust of the Earth [1]. Currently, the PMx and PMy have been observed and estimated accurately by using modern space geodesy techniques, e.g., Lunar Laser Ranging (LLR) [2], Satellite Laser Ranging (SLR) [3], Global Navigation Satellite Systems (GNSS) [4,5], Doppler Orbitography and Radiopositioning Integrated by Satellite (DORIS) [6], and very-long-baseline interferometry (VLBI) [7,8]. The data processing’s complicacy and measurement model’s complexity make it highly time consuming to estimate accurate PMx and PMy. Nevertheless, accurate PMx and PMy are usually only available after hours to days of the observation, which cannot fulfil the requirement of real-time navigation for satellites and spacecrafts [9]. Consequently, it is extremely important to accurately predict the PMx and PMy.

At present, there are many methods that have been studied and used in PMx and PMy predictions, e.g., the Kalman filter [10], the least-squares (LS) collocation method [11], the Kalman filter + predicting atmospheric angular momentum [12], the LS extrapolation + autoregressive (AR) prediction (LS+AR hybrid method) [13,14,15,16,17], the LS extrapolation + spectral analysis method [18], the artificial neural network method (ANN) [19], the wavelets + fuzzy inference systems method [20], modelling and forecasting excitation functions with the Kalman filter [21], the LS+MAR (multivariate autoregressive) hybrid method [22,23], the LS+AR+Kalman hybrid method [24], the singular spectrum analysis (SSA)+LS+ARMA (autoregressive moving average) method [25], and the SSA + Copula-based analysis method [26,27]. With regard to all predicted intervals, the EOP PCC’s (Earth orientation parameters prediction comparison campaign) results show that none of them is superior than the others. Throughout these existing predicted means, the LS+AR hybrid method is regarded as one of the most capable approaches on PMx and PMy predictions [28,29]. In addition, the LS+AR and LS+MAR hybrid methods are also frequently used in the Second Earth Orientation Parameters Prediction Comparison (EOP PCC2) (http://eoppcc.cbk.waw.pl/, accessed on 1 September 2024). Furthermore, Xu et al. show that the PMx and PMy ultra-short-term predictions (1–10 days) of the LS+AR+Kalman hybrid method outperform that of the LS+AR hybrid method [24]. However, for the PMx and PMy ultra-short-term predicted performance requirements of the Global Geodetic Observing System (GGOS) of the International Association of Geodesy (IAG) [30], the current accuracy of the PMx and PMy ultra-short-term predictions still need further improvement by some methods, e.g., by combining these existing predicted means. Between these existing predicted means, the LS+MAR hybrid method has the potential to improve the polar motion ultra-short-term prediction performance [22,31], and the Kalman filter is ideal for real-time parameter estimation and prediction [24], and the LS+MAR hybrid method by using the Kalman filter, namely the LS+MAR+Kalman hybrid method, is worth studying. To the best of our knowledge, a PM prediction performance analysis of the LS+MAR+Kalman hybrid method is still missing.

In this study, we examined the LS+MAR+Kalman hybrid method on PMx and PMy ultra-short-term predictions. To this end, first, the methods of LS+AR, LS+AR+Kalman, LS+MAR, and LS+MAR+Kalman are described. Then, the PMx and PMy ultra-short-term predictions are implemented and analyzed. Finally, the results are summarized.

## 2. Materials and Methods

In this study, the eventual PMx and PMy predictions are performed by the composite of the periodic and linear extrapolation (processed with the LS method) and the residual prediction (processed with the MAR method). This study explored the combination of LS and MAR by using Kalman filter for predicting PMx and PMy.

### 2.1. Least-Squares

For the linear and periodic parts of PMx and PMy, their fitted and extrapolated equation modelled by LS can be expressed as below:(1)$$f_{t}=a+bt+\sum_{j=1}^{k}\left[c_{j}\cos\left(2\pi t/PE_{j}\right)+d_{j}\sin\left(2\pi t/PE_{j}\right)\right]$$
(2)$$\{\begin{array}{c} X^{LS}={\left[\begin{array}{cccc} a & b & c_{1} & \begin{array}{ccc} d_{1} & \cdots & \begin{array}{cc} c_{k} & d_{k} \end{array} \end{array} \end{array}\right]}^{T}\\ A_{t}^{LS}={\left[\begin{array}{c} 1\\ t\\ \cos\left(2\pi t/PE_{1}\right)\\ \sin\left(2\pi t/PE_{1}\right)\\ \vdots \\ \cos\left(2\pi t/PE_{k}\right)\\ \sin\left(2\pi t/PE_{k}\right) \end{array}\right]}^{T}\\ L^{LS}={\left[\begin{array}{ccc} f_{1} & f_{2} & \begin{array}{cc} \cdots & f_{t} \end{array} \end{array}\right]}^{T} \end{array}$$
where $L^{LS}$ is the basic observation data time series, $f_{t}$ is the PM at time t, k is the periodic number, and $PE_{j}$ are the corresponding periodic values, and they are 435.00 and 365.24 in this study. T is the transpose operation. $A_{t}^{LS}$ is the corresponding observation coefficient, and $X^{LS}$ are the parameters that need to be estimated.

### 2.2. AR Model

For the LS fitted residuals, their equation modelled with the AR(p) method is written as follows:(3)$$Z_{t}=\sum_{i=1}^{p}\varphi_{i}Z_{t-i}+\varepsilon$$
(4)$$\{\begin{array}{c} X^{\mathrm{AR}}={\left[\begin{array}{cccc} \varphi_{1} & \varphi_{2} & \cdots & \varphi_{p} \end{array}\right]}^{T}\\ A^{AR}=\left[\begin{array}{cccc} Z_{1} & Z_{2} & \ldots & Z_{p}\\ Z_{2} & Z_{3} & \ldots & Z_{p+1}\\ \vdots & \vdots & \ddots & \vdots \\ Z_{t} & Z_{t+1} & \cdots & Z_{p+t-1} \end{array}\right]\\ L^{AR}={\left[\begin{array}{ccc} Z_{p+1} & Z_{p+2} & \begin{array}{cc} \cdots & Z_{p+t} \end{array} \end{array}\right]}^{T} \end{array}$$
where $X^{\mathrm{AR}}$ are the unknown parameters, $Z_{t}$ refers to the LS fitted residual, ε is the white noise with zero mean, and p is the AR’s order estimated using the BIC (Bayes information criterion) [32].

### 2.3. MAR Model

In addition, for the LS fitted residuals, their equation modelled with the MAR(p) method is written as follows:(5)$$Y_{t}=A_{1}^{Z}Y_{t-1}+\cdots +A_{p}^{Z}\;Y_{t-p}+\varepsilon$$
(6)$$L^{MAR}={\left[\begin{array}{cccccccc} Y_{p+1}^{1} & Y_{p+1}^{2} & Y_{p+2}^{1} & Y_{p+2}^{2} & \cdots & \cdots & Y_{N}^{1} & Y_{N}^{2} \end{array}\right]}^{T}$$
(7)$$\begin{array}{l} X^{MAR}={\left[A_{1}^{Z1}A_{1}^{Z2}\cdots A_{p}^{Z1}A_{p}^{Z2}\right]}^{T}=\\ {\left[\begin{array}{cccccccc} a_{1,1}^{1} & a_{1,1}^{2} & a_{2,1}^{1} & a_{2,1}^{2} & \cdots & \cdots & a_{1,p}^{1}a_{1,p}^{2}\;a_{2,p}^{1} & a_{2,p}^{2} \end{array}\right]}^{T} \end{array}$$
(8)AMAR=[Yp1Yp200  Y11Y120000Yp1Yp2⋯ 00Y11Y12Yp+11Yp+1200⋯ Y21Y220000Yp+11Yp+12⋯ 00Y21Y22⋮⋮⋮⋮⋱ ⋮⋮⋮⋮          YN−11YN−1200  YN−p1YN−p20000YN−11YN−12  00YN−p1YN−p2]
where $Y_{t}={\left(\begin{array}{cc} Y_{t}^{1} & Y_{t}^{2} \end{array}\right)}^{T}$ is a random vector of PMx and PMy, $X^{MAR}$ is the unknown parameters, and the MAR’s order can also be estimated based on the BIC.

### 2.4. Kalman Filter

As we know, the observation equation and state equations of the Kalman filter can be written as follows [33]:(9)$$\{\begin{array}{c} L_{t}=B_{t}X_{t}+\Delta_{t}\\ X_{t}=\phi_{t,t-1}X_{t-1}+\Gamma_{t,t-1}\Omega_{t-1} \end{array}$$
where $L_{t}$ refers to the observation at time t, $B_{t}$ refers to the coefficient of observation, $X_{t}$ refers to the state of the parameters that need to be estimated, $\Delta_{t}$ refers to the observation noise. In addition, $\phi_{t,t-1}$ refers to the state transition matrix, $\Omega_{t-1}$ refers to the system noise matrix, and $\Gamma_{t,t-1}$ refers to the interference matrix. Therefore, the above equations can be solved as below:(10)$$\{\begin{array}{c} X_{t,t}=X_{t,t-1}+J_{t}\left(L_{t}-B_{t}X_{t,t-1}\right)\Omega_{t}\\ D_{t,t}=\left(I-J_{t}B_{t}\right)D_{t,t-1} \end{array}$$
where
(11)$$\{\begin{array}{c} X_{t,t-1}=\phi_{t,t-1}X_{t-1,t-1}\\ D_{t,t-1}=\phi_{t,t-1}D_{t-1,t-1}\left(X\right)\phi_{t,t-1}^{T}+\Gamma_{t,t-1}D_{t-1,t-1}\left(\Omega \right)\Gamma_{t,t-1}^{T}\\ J_{t}=D_{t,t-1}B_{t}^{T}{\left[B_{t}D_{t,t-1}B_{t}^{T}+D_{t,t}\left(\Delta \right)\right]}^{-1} \end{array}$$

Here, $J_{t}$ refers to the filter gain matrix, I refers to the unit matrix, $X_{t,t-1}$ refers to an estimated value of $X_{t}$, and $D_{t,t}$ refers to the covariance matrix of the error.

### 2.5. AR Model Estimated by Using Kalman Filter

By combining the AR model and Kalman filter, the observation equation of AR model can be rewritten as:(12)$$L_{t}^{\mathrm{AR}}=A_{t}^{\mathrm{AR}}X_{t}^{\mathrm{AR}}+\varepsilon_{t}$$
when using Kalman filter to estimate $X_{t}^{\mathrm{AR}}$ of the AR model, $X_{t}^{AR}=\phi_{t,t-1}X_{t-1}^{AR}$, and $\phi_{t,t-1}=I$. So, the unknown parameters $X_{t}^{\mathrm{AR}}$ are re-estimated based on the Kalman filter Equations (10) and (11).

### 2.6. MAR Model Estimated by Using Kalman Filter

Similarly, by combining MAR model and Kalman filter, the observation equation of MAR model can be re-expressed as below:(13)$$L_{t}^{\mathrm{MAR}}=A_{t}^{\mathrm{MAR}}X_{t}^{\mathrm{MAR}}+\varepsilon_{t}$$
when using Kalman filter to estimate $X_{t}^{\mathrm{MAR}}$ of the MAR model, $X_{t}^{MAR}=\phi_{t,t-1}X_{t-1}^{MAR}$, and $\phi_{t,t-1}=I$. Then, the unknown parameters $X_{t}^{\mathrm{MAR}}$ are re-estimated based on the Kalman filter Equations (10) and (11).

### 2.7. Predicted Error Analysis

Being the same as the quality parameter adopted by the EOP PCC [28], we also use the MAE (mean absolute error) to assess the performance of the above predicted methods, and the MAE [33] is written as follows:(14)$$\left(MAE\right)=\frac{1}{N}\sum_{i=1}^{N}\left|O_{i}-P_{i}\right|$$
where $P_{i}$ refers to the i-th predictive PMx or PMy, $O_{i}$ refers to the corresponding observed PMx or PMy, which is the PMx or PMy of IERS (International Earth Rotation and Reference Systems Service) EOP 14 C04 [9] in here, N refers to the total number of prediction, and $O_{i}-P_{i}$ refers to the error in the i-th prediction.

## 3. Results

Four cases were used to test the PM ultra-short-term prediction (1–10 days) performance of the proposed LS+MAR+Kalman hybrid method, and they were Cases LS+AR, LS+AR+Kalman, LS+MAR, and LS+MAR+Kalman, representing the LS+AR hybrid method, LS+AR+Kalman hybrid method, LS+MAR hybrid method, and LS+MAR+Kalman hybrid method, respectively.

### 3.1. Data Description

The IERS EOP 14 C04 product from 1 January 2010 (MJD: 55197) to 16 May 2021 (MJD: 59350) was selected as the test data set, and the PM from 19 October 2020 (MJD: 59141) to 16 May 2021 (MJD: 59350) was predicted. The N of Equation (14) is equal to 210. For the 10-day-ahead prediction, each prediction was performed iteratively by days. In addition, the experiment was made with the IERS EOP 14 C04 bulletins saved for the dates of their production.

### 3.2. Experiments

The PM ultra-short-term predicted results of Cases LS+AR, LS+AR+Kalman, LS+MAR, and LS+MAR+Kalman are shown in Figure 1. The MAE and the relevant statistics of these four cases are given in Table 1.

As Figure 1 shows, the PMx ultra-short-term prediction performance of LS+AR+Kalman is better than that of other three methods, and the PMy ultra-short-term prediction performance of LS+MAR+Kalman is better than that of other three methods. In detail, for PM ultra-short-term prediction, Table 1 shows that the MAEs of the PMx ultra-short-term predictions of the LS+AR+Kalman hybrid method are 0.24, 0.61, 1.04, 1.49, 1.93, 2.34, 2.70, 3.09, 3.45, and 3.78 mas within 10 days. In addition, the MAEs of the PMy ultra-short-term predictions on each day by using the LS+MAR+Kalman hybrid method are 0.22, 0.47, 0.70, 0.92, 1.16, 1.39, 1.61, 1.81, 1.99, and 2.18 mas. Compared to the MAEs of the PMy ultra-short-term predictions of the LS+AR+Kalman hybrid method, the improvement percentages of that of the LS+MAR+Kalman hybrid method are −2.55, 2.31, 5.09,8.78, 8.87, 9.49, 9.43, 10.14, 12.13, and 12.69%, respectively.

## 4. Discussion

This experiment once again illustrates that no method has the best performance in all ERP prediction and prediction time period [28,29]. From the PM prediction experiments of LS+AR, LS+AR+Kalman, LS+MAR, and LS+MAR+Kalman hybrid methods, the test results show that the X direction predicted with the LS+AR+Kalman hybrid method [24] has the minimum mean absolute error, but the proposed LS+MAR+Kalman hybrid method holds the smallest predicted mean absolute error in the Y direction.

## 5. Conclusions

Improving the PM ultra-short-term prediction is an important issue, which is confirmed by the evidence that accurate PM ultra-short-term prediction is vital to refrain from deviations between the international terrestrial and celestial reference frames. In this study, we proposed the LS+MAR+Kalman hybrid method for PM ultra-short-term prediction.

The above results indicate that the LS+MAR+Kalman hybrid method is able to predict the PMy effectively. In the PMy ultra-short-term prediction, the LS+MAR+Kalman hybrid method not only outperforms the LS+MAR hybrid method, but also reduces the MAEs by 0.14 mas averagely compared to the predictions from the LS+AR+Kalman hybrid method, and the average improvement percentage is 7.64%.

On the other hand, the purpose of this paper is only to improve the PM ultra-short-term prediction performance of the frequently used LS+AR, LS+AR+Kalman, and LS+MAR hybrid methods by proposing the LS+MAR+Kalman hybrid method. In fact, there are still many effective methods, e.g., machine learning (ML)+evolutionary computation (EC) [34]. In the future, it can be further explored how the forecast products generated by these methods can be effectively combined to ultimately provide users with more reliable precise PM ultra-short-term prediction.

## Figures and Tables

**Figure 1 sensors-24-06260-f001:**
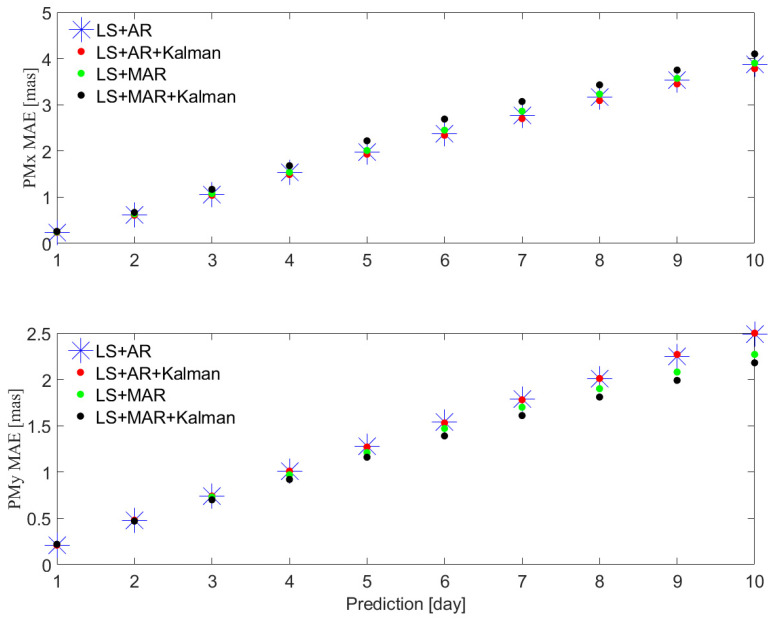
Mean absolute error (MAE) of the four cases for PMx (**top**) and PMy (**bottom**). Blue is the result of Case LS+AR; red is the result of Case LS+AR+Kalman; green is the result of Case LS+MAR; black is the result of Case LS+MAR+Kalman.

**Table 1 sensors-24-06260-t001:** Mean absolute error (MAE) of PMx and PMy ultra-short-term prediction of Cases LS+AR, LS+AR+Kalman, LS+MAR, and LS+MAR+Kalman (units: mas).

Prediction Day/Cases	LS+AR	LS+AR+Kalman	LS+MAR	LS+MAR+Kalman	LS+AR	LS+AR+Kalman	LS+MAR	LS+MAR+Kalman
	PMx	PMy	PMx	PMy	PMx	PMy	PMx	PMy
1	0.24	0.21	0.24	0.21	0.25	0.22	0.26	0.22
2	0.62	0.48	0.61	0.48	0.64	0.47	0.67	0.47
3	1.06	0.74	1.04	0.74	1.08	0.73	1.17	0.70
4	1.53	1.01	1.49	1.01	1.54	0.97	1.68	0.92
5	1.97	1.28	1.93	1.27	2.01	1.21	2.22	1.16
6	2.37	1.54	2.34	1.53	2.45	1.47	2.69	1.39
7	2.77	1.79	2.70	1.78	2.86	1.70	3.07	1.61
8	3.16	2.01	3.09	2.01	3.23	1.90	3.43	1.81
9	3.53	2.25	3.45	2.27	3.57	2.08	3.75	1.99
10	3.87	2.49	3.78	2.50	3.90	2.27	4.10	2.18
Average value	2.11	1.38	2.07	1.38	2.15	1.30	2.30	1.24

## Data Availability

The authors would like to acknowledge the International Earth Rotation and Reference Systems Service (IERS) for providing the related products and support. The PMx and PMy data products from IERS are available at https://www.iers.org/IERS/EN/DataProducts/EarthOrientationData/eop.html, accessed on 1 September 2024.

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
