# Peer review of "Polar Motion Ultra-Short-Term Prediction of Least-Squares+Multivariate Autoregressive Hybrid Method by Using the Kalman Filter"

_sensors, 2024, doi:10.3390/s24196260_

Round 1
Reviewer 1 Report
Comments and Suggestions for Authors
The paper may be of interest for Sensors but it is necessary to answer the following suggestions. Accepted with major revision.
-
It is recommended to provide a more comprehensive comparison with other ultra-short-term polar motion prediction methods to strengthen the article. Including a detailed comparison with different existing approaches in the literature would enhance the understanding of the LS+MAR+Kalman method's performance relative to other methods.
-
To improve the article, it is suggested to enhance the description of the training process for the LS+MAR+Kalman model. Providing more detail and clarity on the training of the model is essential for transparency and reproducibility, thus strengthening the validity of the proposed method.
-
Consider extending the prediction period beyond 10 days to assess the effectiveness of the LS+MAR+Kalman method in reconstructing ultra-short-term polar motion. A longer prediction period would help demonstrate the method's predictive capability and stability over a broader range of time, enhancing the confidence in its performance.
-
To enhance the article's credibility, it is recommended to include evidence or demonstrations showing that the LS+MAR+Kalman method can accurately reconstruct a known period of polar motion. Validating the method with known data sets or historical periods of polar motion would provide a stronger foundation for its effectiveness in real-world applications.
Author Response
Manuscript ID: sensors-3098192
We thank Reviewer 1 very much for comments and suggestions on our manuscript. They are very helpful for improving our manuscript. We carefully addressed all comments of the reviewer and accounted for them in the paper as stated below.

Reviewer 2 Report
Comments and Suggestions for Authors
Review of the article by Zhirong Tan et al on Polar motion ultra-short-term prediction sent to Sensors.
In my opinion, even if the results of the article are correct and English is acceptable, they are explained in an ultra-short manner. Conclusion – two paragraphs saying that prediction is important and hybrid method proposed overperforms others. Such conclusion is typical for most of the narrow prediction papers and can be written just in three words –
– we found the better method!
For comparison see
| Combining evolutionary computation with machine learning technique for improved short-term prediction of UT1-UTC and length-of-day |
| Dhar, Sujata; Heinkelmann, Robert; Belda, Santiago; Modiri, Sadegh; Schuh, Harald; et al. |
| Earth Planets And Space |
Now drawbacks:
Main observation equation 12 L=AX+e is given in 2.4, while it is already needed in 2.1. Coefficients phi_i in 2 becomes Az1 Az2 in 2.3, what is not in agreement with abbreviation A in 2.2. at all ( A there is used for random process Z). In 2.3 X becomes A... and vise versa. Could you unify some how if not the abbreviations - the meaning the matrices and vectors denote in different sections...
AR and MAR models are explained very briefly, section 2.5 is ultra-short, but from it would be interesting to understand, how state vector is estimated, are Euler-Liouville equations involved, what about AAM, OAM excitations, were they somehow used? Or Kalman filter was involved only to remove the observational noises ( which are now less then 0.1 mas?)
AR method, no matter single- or multivariate is usually applied after estimation of autoregression coefficients from autocovariance function.
Kalman filter is usually used to estimate state vector from observations, knowing input excitation and noise covariances. For Kalman filtering transition matrix coefficients (the same as autoregression ones) are supposed to be known from the dynamical system. I mean that AR model is the same differential equation, but in discrete space. How do the authors use this?
If in works of Xu et al the components are separated into LS-modeled, AR-modeled and Kalman filter -derived from AAM, OAM, how they are combined or divided in the presented article is not explained at all.
Why two indexes (t,t) in Xt,t are used? How Dt,t is obtained at the beginning, what is Delta in 11 –
I do not see blues dots in Fig 1 at all. By the way, non-zero MAE for 0-lag means the experiment was made in real-time or with the EOP C04 bulletins saved for the dates of their production. Can the authors write anything on the design of their tests.
In my opinion explanation should be extended to become leaving no questions in a good article.
My solution for present form - reconsider after sufficient revision and extension.
Author Response
Manuscript ID: sensors-3098192
We thank Reviewer 2 very much for comments and suggestions on our manuscript. They are very helpful for improving our manuscript. We carefully addressed all comments of the reviewer and accounted for them in the paper as stated below.

Round 2
Reviewer 1 Report
Comments and Suggestions for Authors
No comments
Reviewer 2 Report
Comments and Suggestions for Authors
Now I understood that you use Kalman filter to estimate AR coefficients.
Then the difference from Xu, Liao, Zhou is only in multivariate AR.
For the next time I would recoomend to make the paper more sound by writing methodology in details. Do not be skimp on explanations.